# The Yin-Yang Concept of Pediatric Obesity and Gut Microbiota

**DOI:** 10.3390/biomedicines10030645

**Published:** 2022-03-10

**Authors:** Lorena Elena Meliț, Cristina Oana Mărginean, Maria Oana Săsăran

**Affiliations:** 1Department of Pediatrics I, George Emil Palade University of Medicine, Pharmacy, Science, and Technology of Târgu Mureș, Gheorghe Marinescu Street No 38, 540136 Târgu Mureș, Romania; lory_chimista89@yahoo.com; 2Department of Pediatrics III, George Emil Palade University of Medicine, Pharmacy, Science, and Technology of Târgu Mureș, Gheorghe Marinescu Street No 38, 540136 Târgu Mureș, Romania; oanam93@yahoo.com

**Keywords:** children, obesity, gut microbiota

## Abstract

The era of pediatric obesity is no longer a myth. Unfortunately, pediatric obesity has reached alarming incidence levels worldwide and the factors that contribute to its development have been intensely studied in multiple recent and emerging studies. Gut microbiota was recently included in the wide spectrum of factors implicated in the determination of obesity, but its role in pediatric obese patients is far from being fully understood. In terms of the infant gut microbiome, multiple factors have been demonstrated to shape its content, including maternal diet and health, type of delivery, feeding patterns, weaning and dietary habits. Nevertheless, the role of the intrauterine environment, such as the placental microbial community, cannot be completely excluded. Most studies have identified *Firmicutes* and *Bacteroidetes* as the most important players related to obesity risk in gut microbiota reflecting an increase of *Firmicutes* and a decrease in *Bacteroidetes* in the context of obesity; however, multiple inconsistencies between studies were recently reported, especially in pediatric populations, and there is a scarcity of studies performed in this age group.

## 1. Introduction

Pediatric obesity has reached epidemic proportions worldwide regardless of age. The prevalence of obesity varies by age, sex, ethnicity, race, and socio-economic status [1]. Obesity prevalence is higher in females of non-Hispanic black and of Hispanic origin when compared to non-Hispanic whites [1,2]. According to the World Health Organization, approximately 40 million children below the age of five years are diagnosed with obesity with particular prevalence in younger children [3]. The global prevalence of pediatric overweight and obesity was reported to vary from 5.7% to 40% depending on the studied population [4,5,6,7]. The multifactorial determination of this condition has been assessed in multiple studies which have shown that obesity occurs in genetically predisposed individuals who fulfill ‘obesogenic’ conditions, such as improper dietary habits, poverty or limited wealth, and even certain cultural traditions [1,8]. Nevertheless, no interdependence between genetic factors and ‘obesogenic’ factors has been established. Without doubt, genetic factors play a major role since multiple studies have shown that certain maternal gene polymorphisms are associated with increased risk for obesity in offspring [9,10,11,12], suggesting that predisposition to the development of obesity begins during intrauterine life. Moreover, the involvement of certain genes in weight gain by regulating feeding behavior, appetite, metabolism and energy expenditure has been clearly demonstrated in multiple studies [8]. Thirty-two loci of the human genome were found to be associated with increased body weight, but only 2% of the population was found to carry these genes [13]. In terms of the exogenous determination of obesity, energy intake that exceeds energy expenditure, along with a wide-spectrum of factors which interact with this process in multiple ways, represent the most important triggers [14]. The long term-persistence of obesity from childhood to adulthood is definitely a hallmark of decreased life expectancy taking into account the wide spectrum of complications related to this nutritional disorder, such as cardiovascular diseases, fatty liver disease or hepatocellular carcinoma [15]. Several human and animal studies have shown that obesity might be considered a major risk factor for carcinogenesis [16,17,18,19]. These findings were also supported by the relationship observed between pediatric obesity and primary liver carcinoma in later adulthood [20]. Most of these complications are directly related to the early inflammatory status identified in overweight and obese children [21]. Subclinical inflammatory status related to pediatric obesity was further demonstrated by the recently observed associations between increased serum levels of leptin and several pro-inflammatory cytokines, such as interleukin (IL) 6, IL 1β and tumor necrosis factor (TNF) α in obese children [22]. It has been suggested that adipose tissue serves not only a storage function, but also a secretory one. Thus, adipokines, when imbalanced, have the ability to stimulate not only a pro-inflammatory state by secreting multiple proteins, such as adiponectin or leptin, as well as cytokines with multiple biological activities, but also an insulin-resistant state, with both contributing to the pathogenesis of non-alcoholic fatty liver disease (NAFLD) and its future progression to non-alcoholic steatohepatitis [23,24,25]. The involvement of immune cells in this pro-inflammatory state is no longer a matter of debate. Macrophages, mast cells, neutrophils, and eosinophils, together with T and B lymphocytes, appear to form a complex array of cells involved in the pathogenesis of obesity [26]. Therefore, adipose tissue acts like a standalone organ, similar to endocrine or immune system organs.

The precise role of each factor involved in obesity etiology remains to be established. It is clear that genetic background and ‘obesogenic’ environment do not fully explain the etiology and variations between obesity prevalence among different populations. Thus, additional factors that might be related to obesity determinism should also be identified and thoroughly studied in order to develop effective strategies to prevent obesity development, and especially its long-term complications. One of these factors seems to be the intestinal microbiota, which has been intensely studied in a large body of emerging literature assuming the hypothesis that changes in its composition are related to both obesity determination and its associated complications [14]. The gut microbiota and humans evolved together and might be considered mutual partners; dysbiosis represents a life-threatening imbalance of this partnership since it has been demonstrated to be associated with several other conditions aside from obesity, such as diabetes, atherosclerosis, allergic and autoimmune disorders, gastrointestinal diseases and neoplasia [27,28,29,30].

## 2. Gut Microbiota and Obesity—To Be or Not to Be?

The gut microbiota might be defined as a shield of protection for mucosal permeability which regulates the fermentation and absorption of dietary polysaccharides, accounting for its role in the regulation of fat accumulation and subsequent obesity development [31]. The gut microbiota and host are in a symbiotic relationship with the gut microbiota contributing to the host’s physiological homeostasis and impairment of the gut microbiota resulting in the development of a wide spectrum of disorders [32]. In terms of obesity, alteration in the gut microbiota leads to an increase in energy harvesting from the diet. Aside from its role in increasing energy harvesting and the regulation of fat storage [32,33], the gut microbiota has also been shown to contribute both to the modulation of substrate formation for storable fat synthesis [34], and the production of compounds that, once absorbed into the systemic circulation, will promote the development of obesity-associated complications by increasing tissue inflammatory injuries and insulin resistance [35].

The gut microbiota might be considered as a true standalone organ with each individual harboring hundreds of different species with only a few dozen conserved between individuals, forming a stable core in a normal healthy state [36]. Based on the definition of a standalone organ, the gut microbiota has multiple functions, including digestion of certain otherwise indigestible dietary fibers, and interaction with several cells, such as immune, metabolic or nervous system cells involved in protection against pathogens, contributing to creation of a barrier against infections [37]. It has long been known that the colonization process begins at birth and is strongly influenced by multiple factors up to the age of three years, such as delivery type, antibiotic administration during early childhood, feeding habits, time of weaning, food composition, and environmental hygiene, which will be discussed further.

Gut microbiota development continues during childhood and adolescence becoming increasingly stable over time, but it is generally accepted that it closely resembles that of an adult by around the age of three years [38,39]. The gastrointestinal tract of a healthy adult has been found to harbor approximately 10^2^ microbial cells within the stomach, duodenum and jejunum, while the distal ileum hosts between 10^7^ and 10^8^ microbial cells, the highest proportion being found in the colon which contains around 10^11^–10^12^ microbial cells [32]. The human gut contains approximately 1–2 kg of microorganisms with >105-fold more genes compared to the human genome itself [32]. The gut microbiome is crucial in maintaining the host’s homeostasis by contributing to the digestive process and energy production, preventing pathogen colonization and regulating the activity of the immune system [40]. Studies have demonstrated that the gut microbiome has the ability to modulate the individual’s susceptibility to obesity by increasing caloric extraction from indigestible dietary fibers as well as by augmenting its storage in adipose tissue [40]. Studies have also revealed that the gut microbiome influences whole body metabolism since germ-free mice displayed alterations in liver, kidney and bowel homeostasis [41,42,43].

Diet is a crucial factor involved in both obesity development and gut microbiota regulation. Thus, it has been demonstrated that dietary patterns shape the appearance of distinct combinations of bacteria within the gut, defined as enterotypes [44], by providing different growth-promoting and growth-inhibiting factors for the development of specific phylotypes [45]. It is well-known that obesity is associated with subclinical inflammation, even in pediatric patients, which may lead to a wide spectrum of long-term complications [21,22]. The gut microbiota seems also to contribute to the development of this inflammatory profile. According to a study performed on overweight and obese subjects, individuals that consumed the most unhealthy food, consisting of confectionary and sugary drinks with low consumption of yoghurt, fruit and water, displayed a significantly more severe inflammatory profile when compared to those with the healthiest diet based on increased consumption of yoghurts, fruits and soups [46]. The same authors concluded that the healthiest dietary pattern was strongly related to the highest microbial diversity within the gut [46]. The complex interaction between obesity and gut microbiota is further supported by other findings that have shown a decrease in microbial community richness in the context of overweight and obesity [47]. As mentioned, not only does obesity seem to be shaped by the gut microbiota, but also obesity-associated inflammation. Thus, increased consumption of fat is associated with a more severe degree of white adipose tissue inflammation and its related complications, while polyunsaturated fatty acid-based diets were found to counteract this inflammatory state in overweight and obesity, promoting a metabolically healthy phenotype [48,49]. In terms of protein consumption, it was shown that, if coupled with exercise, this might exert a positive effect on gut microbiota development. Clarke et al. demonstrated a positive correlation between high protein consumption and gut microbial diversity, but they also showed that athletes with a low body mass index harbor significantly higher levels of *Akkermansia*, which is inversely correlated with obesity in both animals and humans [50,51]. Although the relative abundance of each member of the gut microbial community is highly dynamic during an individual’s lifespan, the overall composition of the gut microbiome can be stable for years [52,53]. Thus, we might assume that host genetic factors might also be involved in shaping the gut microbiota apart from dietary patterns, but the extent of this involvement remains unclear. Studies performed on monozygotic and dizygotic twins failed to demonstrate significant associations in gut microbiota profiles at various ages from infancy to adulthood, suggesting that genetic factors have only a minor role in shaping the human gut microbiota [54,55]. In addition, the influence of dietary patterns on gut microbiota has been found to be sex-dependent with diet affecting gut microbiota development differently in males and females [56]. These findings are in contradiction to the previously mentioned study indicating that host genetic factors remain active participants in shaping gut microbiota, and the precise role of genetic factors requires further clarification.

A wide spectrum of inconsistencies between studies is evident in terms of gut microbiota in obese individuals. Thus, although most studies have reported that obese individuals possess reduced richness of microbes or their genes within the gut [47,54,57,58], decreased diversity was not constant in all obese individuals [47]. Another inconsistency is related to the ratio of *Bacteroidetes* to *Firmicutes* in obese versus lean individuals, which was observed to increase, decrease or not change at all [59]. Ethnicity is also related to these inconsistencies based on two studies that assessed *Prevotella*-dominated communities showing a positive correlation of these communities with body mass index in HIV positive subjects and controls originating from Mexico City [60], but not in older Amish individuals [2]. Another recent study also suggested a role of race/ethnicity in triggering distinct gut microbiota subtypes in the context of obesity [61].

### 2.1. Gut Microbiota Composition and Obesity

The composition of the gut microbiota consists of the phyla *Firmicutes*, *Bacteroidetes*, *Proteobacteria*, *Actinobacteria* and *Verrucomicrobia*, with *Firmicutes* and *Bacteroidetes* accounting for more than 90% of total bacterial species [62]. The gut microbiota in healthy individuals comprises a high ratio of *Bacteroidetes* to *Firmicutes*, while this ratio is reversed in obese individuals with a higher abundance of *Firmicutes* [63]. Moreover, obese individuals were observed to possess a high level of *Lactobacillus* species and a relatively low abundance of *Bacteroides vulgates* [64]. Metagenomic studies have suggested that the ratio of *Bacteroidetes* to *Firmicutes* might be used as a biomarker of obesity susceptibility [54,65] (Figure 1). Metagenomic analysis of the gut microbiome revealed that 40% of the intestine microbial gene pool is common among all humans around the world, indicating the presence of a core microbiome [66]. Obesity metabolic changes were shown to be influenced by an increase in short-chain fatty acid producers (*Actinobacteria*) and pathogenic *Proteobacteria*, with increase in their abundance in the context of obesity [67]. Interestingly, these alterations in gut microbiota were found to be reversible with weight loss [68].

Several mechanisms have been proposed to account for the influence of the gut microbiota on obesity development. One suggested mechanism is related to the ability of microorganisms to ferment dietary polysaccharides otherwise not digested by humans [35]. Thus, the formation process of dietary fibers results in an increased production of short-chain fatty acids (e.g., butyrate, acetate, propionate) identified in fecal or cecal samples, which, after absorption, induce lipogenesis and promote triglyceride storage through various molecular pathways, such as the activation of carbohydrate responsive element-binding protein and sterol regulatory element-binding transcriptor 1 [35,69]. Another peculiarity of short-chain fatty acids is their ability to suppress the fasting-induced adipocyte factor leading to the inhibition of lipoprotein lipase and subsequent accumulation of triglycerides in host adipocytes [35]. Nevertheless, several positive effects of short-chain fatty acids on human metabolism were suggested by Morrison and Preston in a recent review, such as a potential role in maintaining the integrity of the gut mucosa (butyrate), beneficial effects on glucose homeostasis, a role in appetite regulation (propionate), and the regulation of immune system and inflammatory response [70]. Considering the inconsistencies reported in the literature related to this topic, further studies are needed to clarify the precise role of short-chain fatty acids in human homeostasis, especially in terms of lipid metabolism.

### 2.2. Maternal Factors

The gut microbiota shaping process begins in utero and the most important factor that influences fetal wellbeing is maternal diet [71,72]. According to Gibson et al., offspring born to mothers consuming fish oil were found to have a higher level of opportunistic bacteria, such as *Enterococcus faecium*, *Bilophila wadsworthia* and *Bacteroides fragilis*, with a negative impact on the infant’s immune responses [73]. In contrast, the study of Chu et al. [71] found a reduced level of *Bacteroidetes* in infants whose mothers consumed a high-fat diet during pregnancy. Habitual maternal intake of probiotic-containing food leads to a reduced risk for both spontaneous preterm delivery [74], and allergic disorders during childhood [75]. In addition, maternal alcohol consumption during pregnancy was demonstrated to negatively influence both maternal and neonatal gut microbiome, and, in terms of neonatal outcomes, it resulted in an increased risk for infections and other disorders later in life [76,77].

Maternal health during pregnancy is another key factor in infant wellbeing. Several studies aiming to identify the relationship between maternal health and infant wellbeing concluded that the microbial composition of infants born to obese mothers differs markedly from those born to lean mothers due to increased exposure of the offspring to oxidative stress [78,79,80]. It is well-known that infants born to mothers diagnosed with diabetes, or those with an increased body mass index, are more prone to developing diabetes and obesity in later life [81,82,83] due to alterations in the maternal gut microbiome during both pregnancy and lactation [84]. A recent study concluded that the microbiome in overweight or obese mothers is associated with an increased risk for early obesity in infants since early life microbial composition predicts obesity development later in life [85]. Nevertheless, the precise composition of pregnant women’s gut microbiota is not yet clearly established and reported findings remain contradictory. Collado et al. observed increased levels of *Clostridium*, *Bacteroidetes* and *Staphylococcus*, with a decreased level of *Bifidobacterium*, in mothers with overweight/obesity [78], while the study of Santacruz et al. found an abundance of *Staphylococcus*, *Enterobacteriaceae* and *Staphylococcus*, with a significantly lower level of *Bifidobacterium* and *Bacteroides*, when assessing a similar group [86].

Dysbiosis of the placental microbiota might represent another important factor for neonatal health. Despite the fast colonization during the first three years of life, the hypothesis that the fetus is seeded with bacteria during the intrauterine life from the maternal placental microbial community cannot be ruled out [87]. The few studies that have assessed placental microbial composition observed an increase in the order *Pseudomonadales*, and the genera *Actinobacteria* and *Proteobacteria*, along with lower levels of *Bacteroidetes* and *Firmicutes* in pregnant women with gestational diabetes mellitus [88,89]. Furthermore, studies that compared meconium samples of neonate subjects born to mothers with diabetes to those whose mothers were non-diabetic found an abundance of the *Bacteroidetes* and *Lachnospiraceae* families and the *Parabacteriodes* genus along with decreased levels of *Proteobacteria* in meconium samples from the diabetes group [90].

Based on the above, we might hypothesize a complex interaction between maternal diet and obesity, gestational diabetes mellitus, placental microbial composition and the risk of childhood obesity through important alterations in the newborn’s gut microbiome, beginning from intrauterine life.

### 2.3. Delivery Mode

It has been well-documented that the delivery mode shapes the neonatal gut microbiome. Thus, when born vaginally, the neonate comes in contact with the maternal vaginal and gut microbiota and the infant’s gut microbiome will be similar to the vaginal microbial composition, being dominated by the genera *Lactobacillus*, *Sneathia* and *Prevotella* [91]. Otherwise, if the baby is born by cesarean section, the major component of the neonate’s gut microbiome resembles the mother’s skin microbiome and nosocomial surroundings [92]. Recent studies have revealed an important relationship between cesarean section delivery and the risk of obesity during childhood [93,94]. It was found that infants born by cesarean section possess a lower diversity of the gut microbial community compared to vaginally delivered infants, with lower levels of both *Bifidobacteria* and *Escherichia-Shigella*, along with a complete absence of *Bacteroides* [95,96,97,98]. These infants display a gut microbial community mainly composed of the genera *Staphylococcus*, *Propionibacterium* and *Corynebacterium,* resembling the mother’s skin microbial community [91].

### 2.4. Feeding Patterns

Breastfeeding remains the most appropriate alimentation pattern for infants since human milk contains important bioactive compounds, such as oligosaccharides, which are essential for infant development and offer protection against several childhood disorders, such as obesity [99], type 2 diabetes mellitus [100], celiac disease [101], allergies [102], diarrhea [103], and other metabolic diseases [104]. Moreover, human oligosaccharides have the ability to promote *Bifidobacteria* and *Bacteroides* growth [105], and to regulate infant wellbeing due to their action as prebiotics, modulators of innate immune responses and their anti-inflammatory properties [106]. Additional evidence suggests that *Bifidobacterium* species, such as *B. longum*, *B. breve*, *B. infantis* and *B. pseudocatenulatum*, represent the most common Actinobacteria in breastfed infants [107,108,109], along with lactic acid bacteria, such as *Enterococcus*, *Lactobacillus* and *Clostridium* species [110]. Maternal health is another crucial modulator of human milk content resulting in alterations of the human milk microbiome and subsequent negative changes in the infant gut microbiome [92]. Maternal obesity was found to alter the human milk microbiome by decreasing bacterial community diversity, leading to lower levels of *Bifidobacterium* and higher levels of *Staphylococcus* [111,112]. Recent evidence has also underlined the role of human milk in maintaining infant protein balance [113]. In terms of formula-fed infants, a recent study, which assessed the composition of formula-fed and breast-fed infants gut microbiota at the age of 40 days, found that α diversity was lower in a breast-fed group compared to formula-fed infants [114]. These findings were supported by other studies which concluded that the gut microbiota of formula-fed infants is more diverse compared to that of breastfed infants, being similar to that of older children [115,116]. The most predominant genus in both breastfed and formula-fed infants are *Bifidobacterium* and *Enterobacteriaceae* which seem to decrease with age to almost zero by 18 months [114,117]. In addition, formula-fed infants were found to have increased levels of both *Clostridia* and *Veillonella* spp. compared to breastfed infants [108,114,116]. Considering that human milk is incontestably the most suitable feeding pattern for infants, further studies are required to identify the precise role of each bacterium within the gut microbial community and the differences between feeding patterns.

### 2.5. Weaning and Diet

Diet definitely plays a crucial role in modulating the microbial community with it being observed that, once weaned, infants begin to develop a similar microbial composition to that of adults with an increase in diversity and significant changes in taxonomic groups [108,109]. Most changes in gut microbiota composition are observed between the ages of 9 to 18 months, consisting of a decrease of *Lactobacilli*, *Bifidobacteria* and *Enterobacteriaceae* and an increase in *Bacteroides* and *Clostridium* species, reinforcing the major role of weaning in triggering these changes [110]. Moreover, the type of diet contributes to the composition of the gut bacterial community. Assessing the role of dietary habits in children with obesity, Nakayama et al. observed that a high-fat westernized diet results in a decrease in *Prevotellaceae* in the gut microbiota which might contribute to obesity development during childhood [118]. However, the authors stated that children with a carbohydrate-based diet showed an increase in *Prevotellaceae* and a decrease in *Bacteroidaceae* [118]. Therefore, weaning and diet probably represent decisive factors in gut microbiota development since they are the last factors shaping the microbial community before the age of three years, when this process achieves relative stability.

The influence of the above-mentioned factors involved in shaping gut microbiome during early childhood is summarized in Table 1. The precise role of each factor in the development of pediatric obesity is a topic that definitely requires further research (Figure 2).

## 3. Summary

The gut microbiota comprises a wide range of bacteria with complex physiological functions which extend to extraintestinal tissues and seem to be related to excessive weight gain [33]. It has been demonstrated that the gut microbial community might have a dichotomous role influenced by environmental factors, with the ability to promote energy storage or, in contrast, to favor leanness [110,119,120]. The shaping of the gut microbiota begins in utero and continues up to the age of 3–4 years being influenced by the maternal diet, amniotic fluid and placental microbial community, type of delivery, feeding practices and hygiene habits, such as brushing the teeth after meals [121]. The most important players related to obesity risk in the gut microbiota are the *Firmicutes* and *Bacteroides*. Studies on both animal and human subjects have found that obesity is associated with an increase in *Firmicutes* and a decrease in *Bacteroidetes*, independently of dietary caloric intake [34,122]. A recent systematic review [121] supported this statement, highlighting that several studies have found a positive relationship between higher levels of certain microbes belonging to the *Firmicutes* (e.g., *Eubacterium halli*, *Clostridium leptum* and *Lactobacillus* spp.) and overweight or obesity [110,119]. Nevertheless, the findings assessing the relationship between gut microbial composition and obesity remain contradictory, since other studies have found the contrary, with observed decrease in bacteria belonging to the same phyla (e.g., the genera *Staphylococcus* and *Clostridium difficile* [33,120]). Moreover, several studies have found higher levels *of Bacteroides fragilis* in children with overweight/obesity compared to normal-weight children [119,120,123], while others have found that overweight/obesity is associated with a decrease in the entire phylum *Bacteroidetes*, along with a decrease in the *Bacteroides/Prevotella* groups [124,125]. In addition, a meta-analysis that included mostly adults failed to find any association between the *Bacteroidetes/Firmicutes* ratio and obesity [65]. A recent study performed on 3-year-old children found inverse relationships between both *Parabacteroidetes* and *Peptostrecoccae* U and overweight/obesity [126]. The same study found a tendency of overweight/obese children to exhibit relative abundance of *Ruminococcus* and *Akermansia* [126]. Thus, the authors concluded that several differences in gut microbial composition between obese and lean adults are similar to the 3-year-old age group suggesting obesity predisposition has its origins in early childhood [126].

Regardless of the inconsistencies identified between studies, we must acknowledge the major role of gut microbiota in the development of childhood overweight/obesity. Additional studies to clarify the precise role of each bacterium from the gut microbiota would represent a major development in the era of pediatric obesity. Based on the wide spectrum of long-term complications associated with increased body mass index, filling the existing gaps regarding the relationship between gut microbial composition and pediatric obesity might be the cornerstone for the development of innovative prophylactic and/or therapeutic strategies targeted at decreasing the incidence of overweight/obesity in both children and adults.

## Figures and Tables

**Figure 1 biomedicines-10-00645-f001:**
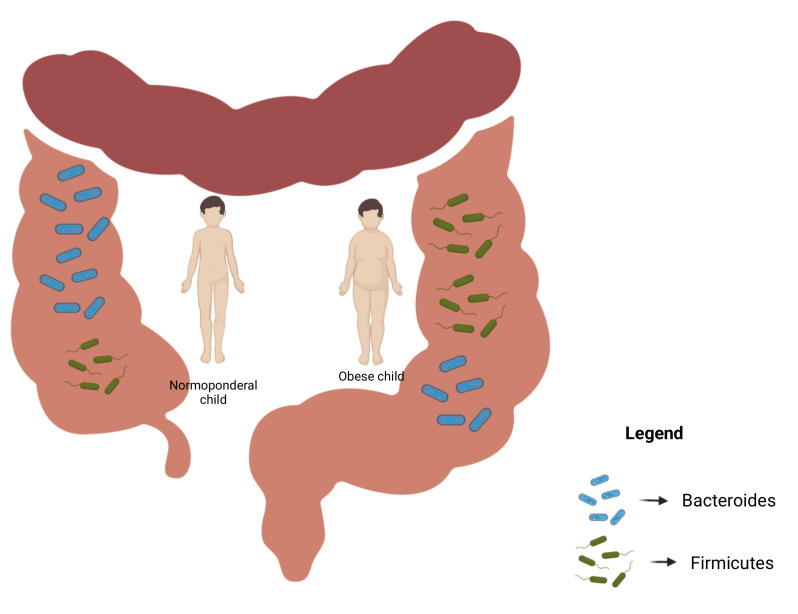
The ratio of Bacteroides/Firmicutes—a biomarker of obesity succeptibility (created with www.BioRender.com, accessed on 6 February 2022).

**Figure 2 biomedicines-10-00645-f002:**
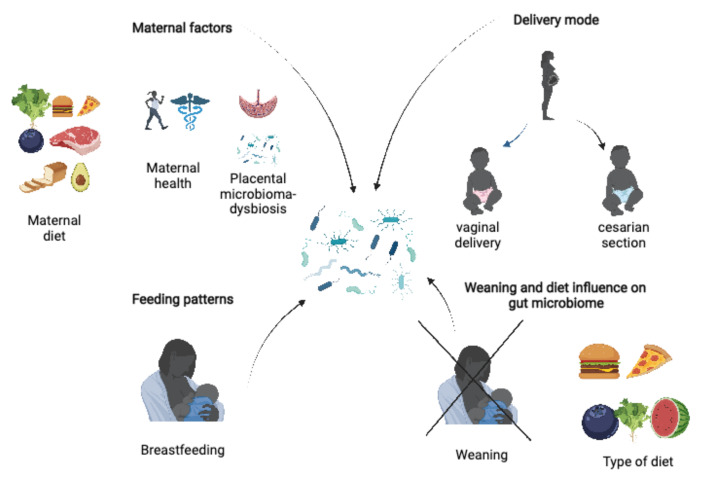
Factors influencing gut microbiome in children (created with www.BioRender.com, accessed on 6 February 2022).

**Table 1 biomedicines-10-00645-t001:** The factors influencing gut microbiome in children with obesity.

Factors			Influence on Infant’s Gut Microbiota, Authors, Year
Maternal factors	Maternal Diet	fish oil consumption	higher level of *Enterococcus faecium*, *Bilophila wadsworthia* and *Bacteroides fragilis—*a negative impact on the infant’s immune responses (Gibson et al., 2016 [73]).
		high-fat diet	reduced level of *Bacteroidetes* (Chu et al., 2016 [71])
		probiotic-containing food	reduced risk of preterm delivery (Myhre et al., 2011 [74]), and allergic disorders during childhood (Bertelsen et al., 2014 [75]).
		alcohol consumption	negative influence on both maternal and neonatal gut microbiome (Dubinkina et al. 2017, Labrecque et al., 2015, [76,77]). increased risk for infections and other disorders later in life (Dubinkina et al. 2017, Labrecque et al., 2015, [76,77]).
	Maternal health	overweight and obesity	increased exposure to oxidative stress (Collado et al., 2010, Galley et al., 2014, Gallardo et al., 2015, [78,79,80]).
		diabetes or increased body mass index	higher risk of developing diabetes and obesity (Deierlein et al., 2011, Mehta et al., 2012, Gaillard et al., 2013, [81,82,83]).
		microbiome in overweight or obese mothers	increased risk for early obesity in infant => their microbial composition predicts obesity development later in life (Stanislawski et al., 2018 [85]).
	Dysbiosis of the placental microbiota	mothers with diabetes	abundance of *Bacteroidetes*, *Lachnospiraceae family* and *Parabacteriodes* and decreased levels of *Proteobacteria* in meconium samples (Hu et al., 2013 [90]).
Delivery mode	Vaginally	direct contact with maternal vaginal and gut microbiota	infant’s gut microbiome is similar to the vaginal microbial composition being dominated by the genera *Lactobacillus*, *Sneathia* and *Prevotella* (Dominguez-Bello et al., 2011 [91]).
	Cesarean section	contact with maternal skin microbiome	neonate’s gut microbiome resembles the mother’s skin microbiome and nosocomial surroundings (Kumbhare et al., 2019 [92]). risk for obesity during childhood (Kuhle et al., 2017, Rutayisire et al., 2016 [93,94]). lower diversity of the gut microbial produce lower levels of both *Bifidobacteria* and *Escherichia-Shigella* and absence of *Bacteroides* (Azad et al., 2013; Rutayisire et al., 2016; Song et al., 2013; Jakobsson et al., 2014 [95,96,97,98]). gut microbial community mainly composed by the genera *Staphylococcus*, *Propionibacterium* and *Corynebacterium* (Dominguez-Bello et al., 2011 [91]).
Feeding patterns	Breastfeeding		promotes *Bifidobacteria* and *Bacteorides* growth (Marcobal et al., 2012 [105]) and regulate infant’s wellbeing modulators of innate immune responses and anti-inflammatory properties (Thurl et al., 2010 [106]). *B. longum*, *B. breve*, *B. infantis* and *B. pseudocatenulatum* (Jost et al., 2012; Bäckhed et al., 2015; Stewart et al., 2018; [107,108,109]) along with lactic acid bacteria, such as *Enterococcus, Lactobacillus* and *Clostridium species* (Bergström et al., 2014 [110]). maternal health is a modulator of human milk content influencing the infant’s gut microbiome (Kumbhare et al., 2019 [92]). maternal obesity alter human milk microbiome determining lower levels of *Bifidobacterium* levels and higher levels of *Staphylococcus* (Cabrera-Rubio et al., 2012; Collado et al., 2012 [111,112]).
	Formula-feeding		diversity of gut microbiota at the age of 40 days was higher than in breast-fed group (Ma et al., 2020 [114]), similar to the gut microbiota of older children (Roger et al., 2010 [115]; Bridgman et al., 2017 [116]). the predominant genus in both breastfed and formula-fed infants are *Bifidobacterium* and *Enterobacteriaceae* which decrease with age (Ma et al., 2020 [114], Yassour et al., 2016 [117] increased levels of both *Clostridia* and *Veillonella* spp. Bäckhed et al., 2015 [108]; Ma et al., 2020 [114]; Bridgman et al., 2017 [116]).
Weaning and diet influence on gut microbiome	Weaning		modulating microbial community determine similar microbial composition to that of adults (Bäckhed et al., 2015; Stewart et al., 2018 [108,109]). at 9 to 18 months will decrease of *Lactobacilli*, *Bifidobacteria* and *Enterobacteriaceae* and an increase in *Bacteroides* and *Clostridium species* (Cabrera-Rubio et al., 2012 [110]).
	Type of diet	children with high-fat westernized diet	decrease in *Prevotellaceae*-obesity development during childhood (Nakayama et al., 2017 [118]).
		children with carbohydrate-based diet	increase in *Prevotellaceae* and a decrease in *Bacteroidaceae* (Nakayama et al., 2017 [118]).

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
