# Peer review of "The Yin-Yang Concept of Pediatric Obesity and Gut Microbiota"

_biomedicines, 2022, doi:10.3390/biomedicines10030645_

Round 1

Reviewer 1 Report

This is an interesting, relevant, and complex topic and as the authors stated, additional studies are necessary to understand the continuously contradictory results published. Due to its complexity, becomes difficult to summarize or find meaningful outcomes that justifies specific occurrences. Therefore, discussing all aspects of obesity and gut microbiota may be challenging. Due to its current relevance, I consider it’s topic worthy of discussing and sharing among the scientific community as a “call” for gathering efforts in better understanding the still existing mysteries behind gut microbiota. The manuscript attempts to show the readers that importance and highlights relevant aspects of the topic that need to be urgently addressed. However, the authors should restructure the main headings into clear and detailed sections without repeating too many times the same information or statements given previously, e.g., in the introduction. Also, final remarks need to be adjusted to what is in fact the final remarks and not a summary (unless it’s called a summary) of the main ideas. I recommend this manuscript’s publication after careful language revisions and sections re-structure. Minor revisions, suggestions and other comments are detailed below.

line 30 -31, this sentence begins with "Thus", a word usually used for consequence of something, which is not the case of this sentence in regards with the previous statement. I would say, for example, "In fact" as the following sentence is showing an example of what was statement. However, this example does not seem to be as essential to include. 

Line

line 42 - misplead word (typo in the word “suggesting”)

Line 45 – the authors refer to multiple studies however only point 1 reference. Or the authors add more references or refer only to this study

Line 51 and 52 – avoid repeating the same word in the same sentence (i.e., disorders)

Line 52 – 53 – improve sentence - suggestion ….Several human and animal studies….

Line 57 – overweighted and obese children

Line 60 – in obese children

Line 189 – “ration” should be ratio

Subsection 2.1 lines referring to SCFA and SCFA producers, there is no mention or examples of what kind of SCFA the studies are referring to. As SCFA have a key role on several health promotion pathways it may confuse the readers to state their less positive effects without clarifying both sides.

Subsection 2.1.1 – There is no need for such heading as subsection 2.1 (Gut microbiota composition and obesity) is per se a heading of the subsequent ones. Heading 2.1.1.1 and so one can fit under heading 2.1 and avoid so many subsections that may “break” the text and ideas flow

Line 216 – remove reference 75 as the following sentence is still referring to the same study. Thus, leave only the last one

Line 263 and 265 – These sentences have the same reference. Same comment as previous

Other comments:

Lines 214 – 228 – The authors mix animal and human studies. Having referenced 1 animal study in mice (ref. 75) and remaining ones to human studies (as it should). I suggest focusing only on human studies as references to closer examples and straight mean of comparison. It gives more strength to the statements. In addition, it should be clearly stated when referring to human or animal studies and avoid any direct comparisons between them.

Many facts pointed out in the subsections of the heading 2.1.1 are already mentioned in the introductions, maybe authors can resume that information in the introduction section as they have a detailed subsection ahead. Thus, avoid repeating same information. For example, information in lines 105 to 110 fits best subsection 2.1.1.1

Subsection 2.1.1.3 (Feeding patterns) in this sections authors only refer to breastfed infant/human milk and no comparison is made with other feeding patterns (e.g., infant formula) as the title suggests. If keeping this, authors should change the heading to e.g., “human milk or breastfed patterns” or add information regarding gut microbiota characteristics in case other feeding patterns applies.

Subsection 2.1.1.4. basically, repeats information given previously regarding diet, gut microbiota diversity and age of establishment, etc. Information must be concise and placed together instead of spread out when there is a specific heading for such statements.

Section 3 – lines 302 -331 are not final remarks but a resume of what was described/stated previously. Final remarks should be a clear and short summary of the main conclusions together with the authors opinion and insights (i.e., 332 to 340) regarding the topic as it is expected to be their expertise.

Author Response

March the 5th 2021

To Editor/Reviewers of Biomedicines,

Dear Editor/Reviewers,

Please find attached a revised version of the manuscript entitled: The yin-yang concept of pediatric obesity and gut microbiotawritten by Lorena Elena Meliț, Cristina Oana Mărginean and Maria Oana Săsăran, Manuscript ID: biomedicines- 1608984.

Firstly, we thank very much the reviewers for their valuable comments and suggestions in order to improve our paper.

Following the reviewers’ concerns and observations, we made some modifications to the initial version of our manuscript, which we described in great detail, according to their recommendations, highlighting them in blue in the attached manuscript as it follows:

Reviewer 1

Comment 1

This is an interesting, relevant, and complex topic and as the authors stated, additional studies are necessary to understand the continuously contradictory results published. Due to its complexity, becomes difficult to summarize or find meaningful outcomes that justifies specific occurrences. Therefore, discussing all aspects of obesity and gut microbiota may be challenging. Due to its current relevance, I consider it’s topic worthy of discussing and sharing among the scientific community as a “call” for gathering efforts in better understanding the still existing mysteries behind gut microbiota. The manuscript attempts to show the readers that importance and highlights relevant aspects of the topic that need to be urgently addressed. However, the authors should restructure the main headings into clear and detailed sections without repeating too many times the same information or statements given previously, e.g., in the introduction. Also, final remarks need to be adjusted to what is in fact the final remarks and not a summary (unless it’s called a summary) of the main ideas. I recommend this manuscript’s publication after careful language revisions and sections re-structure. Minor revisions, suggestions and other comments are detailed below.

Answer 1

Thank you very much for all your valuable time spent on assessing our manuscript and we hope to have fully addressed all of your concerns. We deleted the information from the introduction that were also mentioned in other sections. We also restructured the sections of our manuscript for clarity. We renamed the last section as ‘Summary’ according to your recommendations.

Comment 2

line 30 -31, this sentence begins with "Thus", a word usually used for consequence of something, which is not the case of this sentence in regards with the previous statement. I would say, for example, "In fact" as the following sentence is showing an example of what was statement. However, this example does not seem to be as essential to include. 

Answer 2

Thank you for your suggestion, we replaced the word ‘Thus…’ by ‘In fact’.

Comment 3

Line

line 42 - misplead word (typo in the word “suggesting”)

Answer 3

Comment 4

Line 45 – the authors refer to multiple studies however only point 1 reference. Or the authors add more references or refer only to this study

Answer 4

We corrected our typo error.

Comment 5

Line 51 and 52 – avoid repeating the same word in the same sentence (i.e., disorders)

Answer 5

We apologize for repeating the same word in the same sentence, we replaced ‘cardiovascular disorders’ by ‘cardiovascular diseases’.

Comment 6

Line 52 – 53 – improve sentence - suggestion ….Several human and animal studies….

Answer 6

Thank you for your suggestion. We rephrased the sentence according to your recommendation.

Comment 7

Line 57 – overweighted and obese children

Answer 7

We rephrased according to your recommendation.

Comment 8

Line 60 – in obese children

Answer 8

We rephrased according to your recommendation.

Comment 9

Line 189 – “ration” should be ratio

Answer 9

We corrected our typo error.

Comment 10

Subsection 2.1 lines referring to SCFA and SCFA producers, there is no mention or examples of what kind of SCFA the studies are referring to. As SCFA have a key role on several health promotion pathways it may confuse the readers to state their less positive effects without clarifying both sides.

Answer 10

We introduced a paragraph in order to state also the positive effects of SCFA on human metabolism: ‘Nevertheless, several positive effects of short chain fatty acids on human metabolism were suggested by Morrison and Preston in a recent review such as the potential role in maintaining the integrity of gut mucosa (butyrate), the beneficial effects on glucose homeostasis, the role in appetite regulation (propionate) and the regulation of immune system and inflammatory response[70] Taking into account the inconsistencies reported in the literature related to this topic, further studies are needed for clarifying the precise role of short chain fatty acids on human homeostasis, especially in terms of lipid metabolism.’

We also mentioned the kind of SCFA we mentioned that are involved in lipogenesis and triglycerides storage: ‘…(butyrate, acetate, propionate) identified in fecal or cecal samples…’.

Comment 11

Subsection 2.1.1 – There is no need for such heading as subsection 2.1 (Gut microbiota composition and obesity) is per se a heading of the subsequent ones. Heading 2.1.1.1 and so one can fit under heading 2.1 and avoid so many subsections that may “break” the text and ideas flow

Answer 11

Thank you for your valuable suggestion. We deleted all the unnecessary sections and left only the heading for each subsection according to your recommendations.

Comment 12

Line 216 – remove reference 75 as the following sentence is still referring to the same study. Thus, leave only the last one

Answer 12

We removed the reference 75 according to your and it remained only one.

Comment 13

Line 263 and 265 – These sentences have the same reference. Same comment as previous

Answer 13

Thank you. We removed from the first sentence the reference ‘93’.

Comment 14

Other comments:

 Lines 214 – 228 – The authors mix animal and human studies. Having referenced 1 animal study in mice (ref. 75) and remaining ones to human studies (as it should). I suggest focusing only on human studies as references to closer examples and straight mean of comparison. It gives more strength to the statements. In addition, it should be clearly stated when referring to human or animal studies and avoid any direct comparisons between them.

Answer 14

We removed the study referring to animals (reference 75) according to your suggestion, and we rephrased the following sentence. Moreover, we also revised the manuscript and we stated clearly where we referred to human or animal studies.

Comment 15

Many facts pointed out in the subsections of the heading 2.1.1 are already mentioned in the introductions, maybe authors can resume that information in the introduction section as they have a detailed subsection ahead. Thus, avoid repeating same information. For example, information in lines 105 to 110 fits best subsection 2.1.1.1

 Answer 15

We removed the information that was repeating from the introduction and synthetized in the introduction as it follows: ‘It was long proved that the colonization process begins at birth and it is strongly influenced by multiple factors up to the age of 3 years such as delivery type, antibiotics administration during early childhood, feeding habits, the time of weaning, food composition, and environmental hygiene, which will be further discussed.’

The information that were previously mentioned in the introduction were integrated in each subsection according to their match, by highlighting them in blue.

Comment 16

Subsection 2.1.1.3 (Feeding patterns) in this sections authors only refer to breastfed infant/human milk and no comparison is made with other feeding patterns (e.g., infant formula) as the title suggests. If keeping this, authors should change the heading to e.g., “human milk or breastfed patterns” or add information regarding gut microbiota characteristics in case other feeding patterns applies.

Answer 16

Thank you for your suggestion. We introduced the following paragraph in order to underline the differences between breast-fed and formula-fed infants: ‘In terms of formula-fed infants, a recent study which assessed the composition of formula-fed and breast-fed infants gut microbiota at the age of 40 days underlined that α diversity was lower in breast-fed group as compared to formula-fed infants[114]. These findings were sustained also by other studies which concluded that the gut microbiota of formula-fed infants is more diverse when compared to breastfed infants, being similar to that of older children[115,116]. The most predominant genus in both breastfed and formula-fed infants are Bifidobacterium and Enterobacteriaceae which seem to decrease with age to almost zero by 18 months old[114,117]. In addition, formula-fed infants were found to have increased levels of both Clostridia and Veillonella spp than breastfed infants[108,114,116]. Considering that human milk is incontestably the most suitable feeding pattern for infants, further studies are required to identify the precise role of each bacteria and the differences between feeding patterns.’

Comment 17

Subsection 2.1.1.4. basically, repeats information given previously regarding diet, gut microbiota diversity and age of establishment, etc. Information must be concise and placed together instead of spread out when there is a specific heading for such statements.

Answer 17

It is true that we mentioned several details regarding the same information provided in subsection 2.1.1.4, but they were mainly referring to studies performed in adults, whereas in this subsection we assessed especially studies on children. Therefore, we considered more appropriate to split the information since we wanted to underline the peculiarities regarding this topic in pediatric population.

Comment 18

Section 3 – lines 302 -331 are not final remarks but a resume of what was described/stated previously. Final remarks should be a clear and short summary of the main conclusions together with the authors opinion and insights (i.e., 332 to 340) regarding the topic as it is expected to be their expertise.

Answer 18

Taking into account that at the end of each section we mentioned a statement based on our opinion and insights, we considered more appropriate to change the title of the last section into ‘Summary’ as you previously suggested.

We must also mention that our manuscript was also revised for language errors.

Respectfully,

Prof Cristina Oana Mărginean, MD, PhD

Reviewer 2 Report

The writing quality of this manuscript is very high, and the arrangement of the literature and the discussion of viewpoints are objective, balanced, critical and reflective. This manuscript has a high integration concept and reading value for constructing academic literature (research) results related to intestinal microbiota and childhood obesity. I give this manuscript a very high recommendation for publication.

1. It is recommended to draw a cartoon diagram to illustrate the concept of factors that may affect the intestinal microbiota during the period from the pregnant mother to the three-year-old child.
2. It is recommended to make a table of brief descriptions of the cited documents in the topic "2.1.1.1. ~ -2.1.1.4." to make it easier for readers to read.

Author Response

March the 5th 2021

To Editor/Reviewers of Biomedicines,

Dear Editor/Reviewers,

Please find attached a revised version of the manuscript entitled: The yin-yang concept of pediatric obesity and gut microbiotawritten by Lorena Elena Meliț, Cristina Oana Mărginean and Maria Oana Săsăran, Manuscript ID: biomedicines- 1608984.

Firstly, we thank very much the reviewers for their valuable comments and suggestions in order to improve our paper.

Following the reviewers’ concerns and observations, we made some modifications to the initial version of our manuscript, which we described in great detail, according to their recommendations, highlighting them in blue in the attached manuscript as it follows:

Reviewer 2

Comment 1

The writing quality of this manuscript is very high, and the arrangement of the literature and the discussion of viewpoints are objective, balanced, critical and reflective. This manuscript has a high integration concept and reading value for constructing academic literature (research) results related to intestinal microbiota and childhood obesity. I give this manuscript a very high recommendation for publication.

Answer 1

Thank you for your positive comments and for appreciating our work.

Comment 2

  1. It is recommended to draw a cartoon diagram to illustrate the concept of factors that may affect the intestinal microbiota during the period from the pregnant mother to the three-year-old child.

Answer 2

Thank you for your suggestion. We introduced a figure according to your recommendation.

Comment 3

  1. It is recommended to make a table of brief descriptions of the cited documents in the topic "2.1.1.1. ~ -2.1.1.4." to make it easier for readers to read.

Answer 3

Thank you for your suggestions. We inserted a table according to your recommendations.

We must also mention that our manuscript was also revised for language errors.

Respectfully,

Prof Cristina Oana Mărginean, MD, PhD